# Mosquito Cell-Derived Japanese Encephalitis Virus-Like Particles Induce Specific Humoral and Cellular Immune Responses in Mice

**DOI:** 10.3390/v12030336

**Published:** 2020-03-19

**Authors:** Yu-Hsiu Chang, Der-Jiang Chiao, Yu-Lin Hsu, Chang-Chi Lin, Hsueh-Ling Wu, Pei-Yun Shu, Shu-Fen Chang, Jui-Huan Chang, Szu-Cheng Kuo

**Affiliations:** 1Institute of Preventive Medicine, National Defense Medical Center, Taipei 11490, Taiwan; misakihamano@gmail.com (Y.-H.C.); cdj1228@gmail.com (D.-J.C.); jeremyhsu1203@gmail.com (Y.-L.H.); chalin1@ms38.hinet.net (C.-C.L.); hl1520604@gmail.com (H.-L.W.); ck123king@gmail.com (J.-H.C.); 2Department and Graduate Institute of Microbiology and Immunology, National Defense Medical Center, Taipei 11490, Taiwan; 3Center for Diagnostics and Vaccine Development, Centers for Disease Control, Ministry of Health and Welfare, Taipei 11561, Taiwan; pyshu@cdc.gov.tw (P.-Y.S.); vivi@cdc.gov.tw (S.-F.C.)

**Keywords:** Japanese encephalitis virus, virus-like particles, baculovirus, mosquito

## Abstract

The Japanese encephalitis virus (JEV) is the major cause of an acute encephalitis syndrome in many Asian countries, despite the fact that an effective vaccine has been developed. Virus-like particles (VLPs) are self-assembled multi-subunit protein structures which possess specific epitope antigenicities related to corresponding native viruses. These properties mean that VLPs are considered safe antigens that can be used in clinical applications. In this study, we developed a novel baculovirus/mosquito (BacMos) expression system which potentially enables the scalable production of JEV genotype III (GIII) VLPs (which are secreted from mosquito cells). The mosquito-cell-derived JEV VLPs comprised 30-nm spherical particles as well as precursor membrane protein (prM) and envelope (E) proteins with densities that ranged from 30% to 55% across a sucrose gradient. We used IgM antibody-capture enzyme-linked immunosorbent assays to assess the resemblance between VLPs and authentic virions and thereby characterized the epitope specific antigenicity of VLPs. VLP immunization was found to elicit a specific immune response toward a balanced IgG2a/IgG1 ratio. This response effectively neutralized both JEV GI and GIII and elicited a mixed Th1/Th2 response in mice. This study supports the development of mosquito cell-derived JEV VLPs to serve as candidate vaccines against JEV.

## 1. Introduction

The Japanese encephalitis virus (JEV) is a mosquito-borne arbovirus that belongs to the genus Flavivirus. An estimated 50,000–175,000 cases of JEV occur annually in Asia [1,2]. Potential outbreaks of JEV occur in Africa [3] and Europe [4,5,6]. JEV leads to death in ~20–30% cases, and ~30–50% of JEV survivors suffer serious neurologic, cognitive, or psychiatric complications for years after they recover from the initial infection [5,7]. Despite the fact that a JEV infection is responsible for an annual loss of 709,000 disability-adjusted life years [8], no antiviral interventions to control JEV infection have yet been approved [9]. JEV GIII immunization programs have proven effective in controlling JEV [10]; however, subsequent JEV genotype I (GI) displacement and poor protective efficacy against JEV genotype V (GV) suggest that an improved vaccine is needed [2,11,12]. New classes of JEV vaccines, such as protein subunit vaccines, DNA vaccines, viral vectors, and vaccines based on virus-like particles (VLPs) have yet to undergo extensive pre-clinical testing [13]; however, research on VLP-based vaccines is gaining traction [14,15]. When precursor membrane protein (prM) and envelope (E) flavivirus proteins are expressed together within a cell, they self-assemble into VLPs, which are released into the extracellular environment [16,17,18]. Candidate JEV VLP-based vaccines produced through DNA transfection into mammalian cells or stable cell lines provide effective protection against JEV in animals [19,20,21,22,23]. However, the expression of toxic E glycoproteins could limit the generation of stable cell lines capable of producing higher yields of VLPs in mammalian cells [24,25,26]. Nonetheless, JEV GI VLPs produced from stable mammalian cell lines have been shown to induce immunity against GI and GIII JEV in mice and swine [27]. The conventional baculovirus expression system has been employed to produce large quantities of JEV VLPs in insect cells [28,29]; however, the antigenicity of *Lepidoptera*-derived JEV VLPs has yet to be elucidated. In both mosquitos and vertebrates, arboviruses confer virion-specific properties to the host [30]. The process of glycosylation, which has important influences over biological activity and the conformation, stability, and antigenicity of proteins, is different between vertebrate and invertebrate cells [31]. Vertebrate and invertebrate cells also differ in terms of lipid content in cellular membranes, which results in different viral envelope structures [32,33]. The carbohydrates produced in insect cells are less complex than those produced in mammalian cells [34]. One mosquito-derived VLP resembles the infectious structure of early arboviruses in humans and shows promise as a candidate vaccine or diagnostic antigen. Mosquito cell-derived dengue 2 VLPs produced from stable C6/36 cell clones have been characterized in immunogenicity [35]. However, an understanding of VLP structures requires a characterization in the presence of epitopes and an estimate of their resemblance to authentic virions [36,37,38,39,40]. Researchers have demonstrated the efficacy of baculovirus as a vector for gene delivery into mosquitoes [41]. In this study, BacMos system was employed to produce mosquito cell-derived JEV VLPs. We compared epitope-specific antigenicity in JEV VLPs and virions by employing MAC-ELISA to analyze the binding of well-characterized human sera from normal individuals and from JEV-infected individuals. The immunogenicity of JEV VLPs also was evaluated using a *BALB/c* mouse model.

## 2. Materials and Methods

### 2.1. Cell and Viral Cultures

Mosquito cell lines CCL-125 (*Aedes aegypti*) and C6/36 (*Aedes albopictus*) were cultured in a RPMI1640 medium (GIBCO, Invitrogen, Carlsbad, CA, USA) containing 10% fetal bovine serum and 1× antibiotic-antimycotic solution (GIBCO, Invitrogen). These cell lines were maintained in an incubator under a humidified atmosphere of 5% CO2 at 28 °C. Another mosquito cell line, AP-61 (*Aedes pseudoscutellaris*), was cultured in L-15 medium (GIBCO, Invitrogen) containing 10% fetal bovine serum and 1× antibiotic-antimycotic. This cell line was maintained in an incubator under a humidified atmosphere at 28 °C. Viral titers of KF667316 (GI), KF667284 (GIII) [12], Nakayama (GIII), and JaGAr-01(GIII) strains propagated in C6/36 cells were determined via plaque assays in BHK-12 cells.

### 2.2. Construction of Transfer Vectors

A transfer vector was constructed by replacing the *NheI*-*NotI* region of pBac-b1-EGFP-Rhir-E35 with a synthetic 2.8-kb *NheI*-*NotI* DNA fragment that contained a hr1pag1-JEV precursor membrane protein and envelope (prME) (AAB66485, 105-794th amino acid)-poly A site (sequences shown in Appendix A) [42,43]. The resulting plasmid was identified via sequencing and named pBac-IR-GFP-hr1pag1-JEV-prME-Rhir-E. The recombinant baculovirus was generated in accordance with the protocols described previously [44]. The resulting recombinant baculovirus was named BacMos-JEV-prME.

### 2.3. Immunofluorescence Assay (IFA)

Mosquito cells were transduced using BacMos-JEV-prME at a multiplicity of infection (MOI) of 5 or infected with the JEV Nakayama strain at an MOI of 0.1. At 3 days post transduction (dpt) or 2 days post infection (dpi), cells were fixed and incubated with Mab (6B6C) anti-JEV E (1:100) or anti-JEV prM monoclonal antibodies (Mybiosource) (1:80) for 1 h (h). Cells were subsequently washed three times with PBS and incubated with either Alexa Fluor 488-conjugated secondary antibodies or Alexa Fluor 594-conjugated secondary antibodies for 1 h. After a final wash with PBS, images of cells were captured using an inverted fluorescence microscope.

### 2.4. Western and Dot Immunoblotting Analysis

Mosquito cells were transduced with BacMos-JEV prME at an MOI of 2 or infected with the Nakayama strain of JEV at an MOI of 0.1. Harvested cell lysates were subjected to WB analysis. The harvested culture medium was centrifuged at 12,000 rpm for 5 min (min), and 100 μL samples were applied to nitrocellulose membranes (PROTRAN, Schleicher & Schuell) using a bio-dot microfiltration apparatus. The membranes were then blocked using Tris-buffered saline (TBS; 100 mM Tris, pH 7.4, 100 mM NaCl) containing 5% (*v*/*v*) non-fat dry milk at room temperature under gentle shaking for 1 h. The membranes were subsequently incubated with a 1:2000 dilution of Mab (6B6C) anti-JEV E or a 1:1000 dilution of rabbit anti-JEV prM serum (Genetex, Irvine, Irvine, CA, USA) in TBS buffer. Following this, membranes were washed three times (15 min per wash) in TBS buffer containing 0.1% Tween 20 under shaking at room temperature and incubated with a 1:10,000 dilution of horseradish peroxidase (HRP)-conjugated secondary antibodies (Chemicon, Billerica, MA, USA) at room temperature for 1 h. Membranes were washed three times again. Following the second round of washing, a SuperSignal West Pico PLUS Chemiluminescent Substrate (ThermoFisher Scientific, Waltham, MA, USA) was used to determine the presence of HRP on membranes. Dot blot images were obtained and quantified using an Amersham Imager 600 photometer. Dot integrated density values were calculated by multiplying the median local background value by the dot area and then subtracting the product from the total dot intensity.

### 2.5. Production, Purification, and Analysis of Mosquito Cell-Derived VLPs

We transduced 1 × 108 AP-61 cells in 100 mL culture medium with BacMos-JEV-prME in growth medium at an MOI of 5. Following incubation for 12 h, the cells were washed twice using PBS. Culture medium was harvested and refreshed at 2 and 4 dpi. Following incubation for another 2 days, a third culture harvest was collected. Culture supernatant was passed through a 0.45-μm filter to remove debris, whereupon the VPLs were concentrated using a MidiKros Module Tangential Flow Filtration (TFF) System (300,000 molecular weight cut-off, Spectrum Repligen, Rancho Dominguez, CA, USA). The concentrated VLPs were subsequently loaded onto a 25% sucrose cushion and centrifuged at 250,000× *g* (P28S rotor; Hitachi) for 12 h. The resulting pellets were resuspended in NTE buffer (50 mM Tris–HCl pH 7.4, 100 mM NaCl, 0.1 mM EDTA) and subjected to a sucrose gradient (15–55%). Following another (18 h) round of centrifugation at 250,000× *g* (P28S rotor; Hitachi), 8 fractions were collected (from top to bottom) for dot blot analysis and transmission electron microscopy (TEM) assays. VLP proteins were analyzed using SDS-PAGE, and the Coomassie blue-stained band corresponding to E was analyzed via microcapillary liquid chromatography-tandem mass spectrometry, performed using the QSTAR XL Q-Tof mass spectrometer (AB Sciex). The VLPs were quantified by comparing the band densities of E proteins to the band densities of known quantities of proteins.

### 2.6. TEM Observation

Purified JE VLPs were deposited on a copper grid and negatively stained using 2% uranyl acetate. TEM observation was performed using a Hitachi HT7700 at an accelerating voltage of 100 kV.

### 2.7. Dynamic Light Scattering (DLS)

The filtered culture medium from BacMos-JEV-prME-transduced AP-61 cells was 10-fold concentrated, diafitrated with a 10-fold volume of high salt buffer (PBS and 0.5M NaCl), and then 5-fold concentrated using TFF. The particle size was characterized using a Malvern Zetasizer Nano ZS (Malvern Instruments Ltd., Worcestershire, UK). Measurements were performed at 633 nm with an angle detection of 173° at room temperature. Each 1 mL sample (125 μL 50-fold concentrated sample + 875 μL PBS) was analyzed in a disposable, solvent-resistant micro cuvette (ZEN0040) (Malvern Instruments Ltd., Worcestershire, UK) at room temperature. Each sample underwent two series of 15 runs (whereby the duration of each run was 10 s). The size distribution of particles was calculated using the Zetasizer software suite.

### 2.8. JEV-Specific IgM-Antibody Capture Enzyme-Linked Immunosorbent Assay (MAC-ELISA)

Immunoplates (96 wells, Nunc maxisorp) were coated with Goat anti-human IgM antibodies (Jackson, West Grove, PA, USA) and left to stand at 37 °C for 1 h. Following this, plates were blocked using blocking buffer (PBS and 1% BSA), and 100-fold dilutions of human sera (six sera of JEV infection: 58912, 58911, 58833, 58682, 58655, and 58556; four normal sera: S10700567, S10700568, S10700569, and S10700570) were added to the wells. Wells were subsequently incubated at 37 °C for 30 min. Following this, free antibodies were washed out, 100 μL of JE VLPs from the culture supernatant of BacMos-JEV-prME-transduced AP-61 cells or virions of JEV JaGAr-01 strain (1 × 10^7^ PFU/mL) was added to each well, and all wells were incubated at 37 °C for another 30 min. Following a secondary wash with PBS, 4000-fold dilutions of AP-labeled mouse anti-flavivirus monoclonal antibody (D56-3) were added to the wells, and wells were incubated at 37 °C for 30 min once more [45]. After a final PBS wash, p-nitrophenyl-phosphate (Sigma, St Louis, MO, USA) was added, and wells were left to incubate at 37 °C for 20 min. The OD405 was determined using a Dynatech MR700 microplate reader.

### 2.9. Vaccination

Experiments were performed using *BALB/c* mice, which were handled in strict accordance with good animal practices, as defined by the Council of Agriculture, Executive Yuan, Taiwan R.O.C. Animal protocols were approved by the Institutional Animal Care and Use Committee of the Institute of Preventive Medicine, National Defense Medical Center. To assess the ability of JEV VLPs to induce humoral immunity, *BALB/c* mice (*n* = 5) were subcutaneously (SC) immunized with three doses of 1 or 4 μg VLPs at 5, 7, and 9 weeks of age. Another two groups of mice were SC inoculated with three doses of either IMOJEV (a licensed JE vaccine) or PBS, thereby serving as positive and negative controls, respectively. Each positive control animal received a 100 μL (1/10 the recommended adult dose) of the attenuated vaccine, administered via SC injection. Blood was collected from mice at 11 weeks of age. Following coagulation, blood was centrifuged and sera was harvested, aliquoted, and stored at −80 °C until use. The cellular immune responses elicited by JEV VLPs, *BALB/c* (*n* = 5), were assessed by SC immunizing mice with two doses of 0.0625, 0.25, 1, or 2 μg VLPs at 5 and 7 weeks of age. To detect IFN-γ-and IL-4-secreting cells, enzyme-linked immunospot (ELISpot) assays were finally performed on splenocytes from one immunized mice of each group (collected 3 months after the final immunization).

### 2.10. Evaluation of IgG, IgG1, and IgG2a Responses Using ELISA Assay

To create pet32C-JEVE, a partial JEV E gene (AAB66485, 106-794th amino acid) from BacMos-JEV-prME was subjected to PCR amplification (forward primer: 5′-CGACAAGGCCATGGGATTCAACTGCCTGGGTATGG-3′; reverse primer: GGTGGTGCTCGAGTGCGGCCCAGGGTAGTGCTGAAGGC) and subcloned into EcoRV and NotI sites of pet32C using the NEBuilder^®^ HiFi DNA Assembly Cloning Kit. The sequence of pet32C-JEVE was confirmed by DNA sequencing. Recombinant protein expression and purification were performed in accordance with methods described in a previous report [44]. In brief, we coated 96-well plates with purified recombinant JEV E protein (2 μg/mL per well) in 100 μL of carbonate-bicarbonate buffer (NaHCO3, 18.2 mM Na2CO3, pH 9.6). Wells were then held at 4 °C overnight. After blocking, 100 μL of two-fold serial dilutions of serum sample (from 1:100) was added to each well and incubated at 4 °C overnight. Following this, 100 μL of diluted horseradish peroxidase (HRP)-conjugated goat anti-mouse IgG (1:5000, KPL) was added to each well, and color was generated by adding a substrate solution of SureBlue TMB peroxidase. Absorbance at 590 nm was measured using a microplate reader. The endpoint titer was defined as the highest serum dilution at which the OD590 value was ≥0.2.

### 2.11. A 50% Plaque Reduction Neutralization Test (PRNT50) Assay

To determine the titers of neutralizing antibodies (Nabs) from immunized mice (3 mice from each group), sera were first diluted two-fold (starting at a concentration of 1:10) in RPMI medium. Following the heat inactivation of the complement, 100 μL of the serial dilutions were incubated with an equal volume of JEV solution containing approximately 100 PFU of JEV I or JEV III. The mixtures were then added to BHK-21 cells (in 12-well plates) and incubated at 37 °C for 1 h. Cells were subsequently cultured with RPMI containing 2% FBS and 0.3% agarose for 72 h. After culturing, plaques were stained and counted. Nabs titers were defined as the reciprocal of the maximum dilution of serum that reduced virus titer by 50%.

### 2.12. Cytokine Assays Using ELISPOT

Three months after the final vaccination, splenocytes (from 1 mouse per group) were isolated to assess the generation of interferon (IFN)-γ and interleukin (IL)-4 using the enzyme-linked immunospot assay (ELISPOT), in accordance with the manufacturer’s instructions (R&D Systems). Briefly, 5 × 105 splenocytes (in triplicate) were added to 96-well plates coated with anti-mouse IFN-γ or IL-4 captured antibodies. The cells were then incubated with JEV VLPs (5 μg/well), ConA (positive control), or PBS (negative control) in RPMI 1640 medium at 37 °C for 48 h. Following incubation with biotin-conjugated Abs (for IFN-γ and IL-4) and streptavidin-AP, single cytokine-positive cells were visualized (by adding corresponding substrates) and counted using a dissection microscope.

### 2.13. Statistical Analysis

All statistical analyses were performed using the GraphPad Prism software version 6.0.1. Differences between the mock group and the test groups were assessed using multiple *t*-tests. *P*-values are mean with SD for *n* = 5 or 3.

## 3. Results

### 3.1. Expression of JEV Viral Structural Proteins in Mosquito Cells

We developed an efficient gene delivery method to facilitate the production of JEV VLPs in mosquito cells. Specifically, we employed a recombinant baculovirus (BacMos-JEV-prME) containing JEV genotype III prME genes driven by a specific promoter (hr1-pag1) to deliver JEV prME genes into mosquito cells and thereby induce protein expression and the secretion of VLPs (Figure 1). IFA results (Figure 2A) revealed intense JEV E glycoprotein signals in all BacMos-transduced or JEV-infected mosquito cells, except for infected CCL-125 cells. IFA signals indicated that the level of JEV E glycoprotein expression in transduced-cells was comparable to that in infected-cells. The high-intensity JEV E glycoprotein signals observed in transduced and infected mosquito cells indicated that JEV E expression levels in BacMos-JEV-prME-transduced mosquito cells were comparable to those in JEV-infected mosquito cells. We assessed the intracellular localization of expressed E glycoproteins by examining infected or transduced mosquito cells using immunofluorescence microscopy. As shown in Figure 2A, E glycoprotein signals were observed exclusively in the cytoplasm of transduced and infected cells. As shown in Figure 2B, IFA also revealed a signal specific to prM in JEV-infected cells and in BacMos-JEV-prME-transduced C6/36 cells. Western blot results (Figure 2C) revealed the accumulation of E glycoproteins in transduced mosquito cells following incubation. The secretion of JEV E glycoproteins in BacMos-transduced mosquito cells was detected by dot immunoblotting assays on culture media harvested at various time points post-transduction. Dot blot analysis results (Figure 3A) showed that three transduced cell lines and one infected cell had secreted JEV E glycoproteins. Specifically, a strong E glycoprotein signal was detected in transduced AP-61 cells at 1 to 6 dpi. A moderate E glycoprotein signal was detected in transduced C6/36 cells at 2 to 6 dpi. However, only a weak E glycoprotein signal was detected in transduced CCL-125 cells. Interestingly, the average intensity of the E glycoprotein signal was stronger for BacMos-JEV-prME-transduced AP-61 cells than for infected AP-61 cells at 1 to 6 dpi (Figure 3B). From this, we can deduce that expression and secretion of JE E glycoproteins were abundant in BacMos-JEV-prME-transduced AP-61 cells.

### 3.2. Characterization of Mosquito Cell-Derived JEV VLPs

The secretion of JEV E glycoprotein was characterized by subjecting concentrated culture medium from transduced-AP-61 cells to centrifugation using a sucrose gradient. Sucrose gradient banding (Figure 4A, right column) revealed a distinct band in the center area of the tube. Dot blot analysis (Figure 4A, left and middle columns) performed on that band revealed that JEV E glycoprotein co-fractionated with prM, which was distributed across the sucrose gradient from fraction 4 to 8 (30 to 55% sucrose). Electron microscopy analysis (Figure 4B) of purified and negatively-stained fractions rich in JEV E glycoproteins revealed the presence of JEV VLPs, which appeared as spherical particles with an average diameter of 30 nm. These measurements were confirmed by DLS analysis (Figure 4B), and the presence of E proteins in VLPs was confirmed by mass spectrometry analysis. Proteins obtained from fraction 5 of the sucrose gradient were separated using SDS-PAGE, whereupon a Coomassie blue-stained band estimated to be in the size range of JEV E (not shown) was excised from the gel and subjected to LC–MS/MS analysis. Multiple peptides specific to E proteins (29% coverage) were identified (Figure 4C). Taken together, these results indicate that JEV VLPs were secreted from BacMos-JEV-prME-transduced mosquito cells.

### 3.3. VLPs Exhibit Virion-Like Epitopes

Sera from JEV-infected individuals were shown to contain antibodies that bind immune-dominant epitopes to virions. As a tracing antibody, cross-reactive mAb D56-3 provided high affinity at conformation sensitive epitopes. MAC-ELISA results (Figure 5) revealed that VLPs and virions exhibited (1) binding to immunoglobulins in sera of all JEV patients and (2) minor binding activity to normal sera. Similarities in MAC-ELISA patterns between capture antigens from VLPs and virions indicated that mosquito cell-derived JE VLPs presented epitope exposure equivalent to that of authentic virions. Overall, these findings demonstrated that mosquito cell-derived JEV VLPs displayed authentic antigens.

### 3.4. Immunization of JEV VLPs Elicited Robust Immune Responses in Mice

Immunogenic properties of JEV VLPs were characterized by immunizing *BALB/c* mice with three doses of VLPs, IMOJEV, or PBS. ELISA assay results (Figure 6A) showed that mice immunized with VLPs produced specific IgG, IgG1, and IgG2a antibodies at significantly higher levels than mice immunized with IMOJEV. Similar levels of IgG1 and IgG2a indicated that a balanced IgG1/IgG2a response was induced in both groups. Nabs test results (Figure 6B) showed that, in all VLP-immunized mice, PRNT_50_ titers against JEV GI and GIII ranged from 10 to 320. However, we did not observe any significant differences in Nabs responses among JEV GI and GIII. We also did not observe significant differences in Nabs titer results between the VLP- and IMOJEV-immunization groups. This is an indication that mosquito cell-derived JEV VLP immunization cross-protects against infections of JEV GI and JEV GIII. Intriguingly, the titers of neutralizing antibodies in the group immunized with 1 μg VLPs were slightly higher than those of mice immunized with IMOJEV. This indicates that the VLPs elicited JEV E-specific IgG antibodies as well as neutralizing antibodies against both JEV G1 and JEV GIII infections. To confirm the type of cellular immune-responses induced by JEV VLP vaccination, we immunized *BALB/c* mice with two doses of VLPs, IMOJEV, or PBS (0.0625, 0.25, 1, or 2 μg). As shown in Figure 6C, ELISpot assay for the detection of IFN-γ-and IL-4-secreting cells was performed using splenocytes of immunized mice three months after the final immunization. Negligible Th responses were observed in spleen cells from both the PBS and IMOJEV groups (without stimulation). Contrary to expectations, we observed strong background signals in ELISpot assays following immunization with VLPs. Accordingly, the densities of reactive spots induced by mice vaccinated with JEV VLPs were similar in terms of specific-IFN-γ response (300–350 spots/2 × 10^5^ cells) and specific-IL-4 response (300–350 spots/2 × 10^5^ cells), indicating that the Th1/Th2 type response was balanced. However, splenocytes from mice immunized with JEV VLPs presented stronger IFN-γ-and IL-4-secreting spots (300–350 spots/2 × 10^5^ cells) than splenocytes from the positive IMOJEV group (30–50 spots/2 × 10^5^ cells). This indicates that the cellular immune response which resulted from JEV VLP immunization was stronger than that which resulted from IMOJEV immunization. Overall, immunization with mosquito cell-derived JE VLPs was shown to induce specific humoral and cellular immune responses in mice.

## 4. Discussion

Despite the effectiveness of vaccinations, arboviruses, such as the Japanese encephalitis virus, pose a serious threat to public health [9]. There is little doubt that advances in JEV vaccines could provide better countermeasures for the JEV infections that continue to occur sporadically. VLPs, which lack viral genetic material, comprise self-assembled multi-subunit protein structures that are configurative and antigenic to corresponding native viruses [46]. VLPs activate the innate immune system, thereby promoting adaptive immunity [47,48,49,50]. Importantly, repeated high-density displays of viral antigens in authentic conformations promote B-cell activation, which result in a strong humoral immune response and enhanced cellular mediated immunity [51,52]. This makes VLPs highly effective subunit vaccines capable of eliciting a strong immune response. Furthermore, the sophistication of current VLP production methods renders many of the containment procedures that are typically required to produce live virus vaccines unnecessary [53]. Overall, VLPs comprise a promising alternative approach for the development of safe, efficacious vaccines and diagnostic antigens. VLP-based vaccines have proven highly effective against a wide range of viral diseases, such as Hepatitis B (Sci-B-Vac™) and human papilloma virus (Gardisil) [50]. This current research demonstrates the efficacy of BacMos in promoting the expression and secretion of JEV E glycoproteins in three mosquito cell lines: C6/36, CCL-125, and AP-61 (Figure 2A and Figure 3), with AP-61 cells providing the highest E glycoprotein yield. Interestingly, the JEV E glycoprotein yield of transduced-AP-61 cells actually exceeded that associated with authentic JEV infection (Figure 3). The process of flavivirus maturation is regulated by the proteolytic cleavage of the precursor membrane protein (prM) [54]. The maturity of VLPs affects the display and antigenicity of epitopes [55]. The conformationally dependent Mab 6B6C-1 specifically binds mature flavivirus VLPs, such that dot blot analysis results obtained using 6B6C-1 Mab (Figure 3A and Figure 4A) indicated that the JEV VLPs were mature. Mosquito cell-derived JEV VLPs presented a density range of ~35–45% sucrose (Figure 4A). TEM and DLS analysis revealed that the average diameter of the JEV VLPs was roughly 30 nm (Figure 4C), which is similar to the size of Zika [56] and JE GV VLPs [23,29] produced using a conventional baculovirus system or stable cell lines.

The maturity of flavivirus VLPs affects epitope display and antigenicity. The 6B6C-1 Mab of the conformational-dependent Mabs performs the specific ability to bind mature flavivirus VLPs [55]. In dot blot analysis (Figure 3) and MAC-ELISA assays (Figure 5), the specific binding of JEV human sera and two conformation-sensitive mAbs (6B6C and D56-3) to VLPs demonstrated that the secreted JEV VLPs maintained virion-like epitopes with maturity. *Flavivirus* VLPs with varying conformations and properties are secreted under different expression systems and culture conditions. For example, in mammalian cells, DEN2 VLPs produced at a lower temperature (31 °C) resembled native dengue viruses. These VLPs also elicited the highest titers of neutralizing antibodies, far exceeding those elicited by VLPs produced at 37 °C [57]. The BacMos system was therefore especially conducive to the production of mosquito cell-derived VLPs at lower temperatures. Based on the fact that humans are dead-end hosts for JEV, the World Health Organization (WHO) has recommended that the JEV-specific MAC-ELISA be employed as the first-line serological assay in the diagnosis of JEV infections [58]. Non-infectious VLPs may provide an alternative capture antigen for JEV-specific MAC-ELISA, which have excellent safety and specificity. Furthermore, in downstream processing, the lytic nature of the conventional baculovirus system complicates the process of purification following VLP formation [59]. Conversely, the lack of process-related contaminants associated with non-lytic BacMos should simplify the downstream purification of VLPs [53].

IMOJEV is comprised of a recombinant chimeric virus and is currently considered the most effective JEV vaccine [60]. A single dose of the IMOJEV vaccine has been shown to protect mice from both the Murray Valley encephalitis virus and the West Nile virus [61]. However, IMOJEV is not recommended as vaccination for pregnant and breastfeeding women as well as immunocompromised persons [62]. Furthermore, due to serological interference, differentiating between the serological responses of yellow fever infection and the effects of IMOJEV vaccination can be difficult. Higher titers of JEV Nabs have been correlated with superior protection and/or good disease outcomes in animal models [63,64,65] as well as in human models [66,67,68,69]. In the current study, JEV VLP immunization resulted in a robust JEV E-specific IgG response, a balanced IgG2a/IgG1 ratio, and the formation of neutralizing antibodies (PRNT_50_ titers with values of 10–320) against JEV G1 and GIII (Figure 6A,B). Contrary to our expectations, lower titers of JEV E-specific IgG were detected in the positive IMOJEV group. Cellular immunity has been shown to play a role in partial protection and recovery from JEV infection in mice [70,71]. Immunization with candidate DNA vaccines which encode JEV envelope glycoproteins has also been shown to induce protective cell-mediated immunity against intracerebral viral challenges [72]. Moreover, the presence of L3T4+ T cells in adoptively transferred effector populations has been found to increase the formation of neutralizing antibodies which protect against JEV infection [73]. In ELISpot assays conducted after VLP immunization, similarities in the reactivity of IFN-γ (Th1 profile) and IL-4 (Th2 profile) confirmed that the Th1/Th2 (IgG2a/IgG1) response was balanced (Figure 6C). JEV VLP immunization was also shown to elicit a large number of IFN-γ-and IL-4-secretion spots, exceeding the effects which were observed in the positive IMOJEV group. The enhanced IFN-γ and IL-4 induced secretion of splenocytes is associated with the functional cytotoxic T cells which are responsible for cytolytic destruction of virus-infected cells. In this study, ELISpot assay has been performed at one week, one month, or two months after the final immunization. However, strong background signals (without VLP stimulation) interfere assays for specific IFN-γ- and IL-4 responses, which results in a major limitation of this study relating to a low number of mice employed in ELISpot assays. Nevertheless, the immune responses induced by VLP immunization in mice were more pronounced than those induced by IMOJEV vaccination in terms of specific IgG antibodies, neutralizing antibodies (1 μg dose), and cellular immunity. However, there are no statistic differences in E-specific IgG response (1 vs. 4 µg doses) (Figure 6A), which indicates that immunization of 1 µg dose could be sufficient for inducing a plateau IgG response. There are also no statistic differences in Nabs (1 vs. 4 µg doses) (Figure 6B) and Th1/Th2 (1 vs. 2 µg doses) (Figure 6C) responses. However, a 1:10 dilution in a 50%PRNT is considered to be indicative of a protection from JEV infection [74]. This indicated the 1 μg dose should be sufficient to maximize the Nabs reaction in mice. Antigen-presenting cells, including dendritic cells and macrophages, are primary targets for JEV infection. The fact that mosquito cell-derived JEV VLPs are structurally similar to native virions could be an advantage to antigen-presenting cells during the recognition process and may actually enhance the efficacy of the vaccine. A mice challenge assay will be conducted to demonstrate the protection effect of VLPs immunization in future research. In the drawback of mosquito cell-derived VLPs, mosquito cell culture fluids containing allergens trigger human hypersensitivity [75]. Thus, the high purity of mosquito cell-derived VLPs will ensure an allergens-contamination reduction for the prevention of an anaphylactic reaction. Overall, this study demonstrated that mosquito cell-derived JEV VLPs represent an effective alternative vaccine against JEV. Our results also demonstrated that the proposed BacMos method has a potential for the scalable production of mosquito cell-derived JEV VLPs as well as of other arbovirus VLPs.

## Figures and Tables

**Figure 1 viruses-12-00336-f001:**
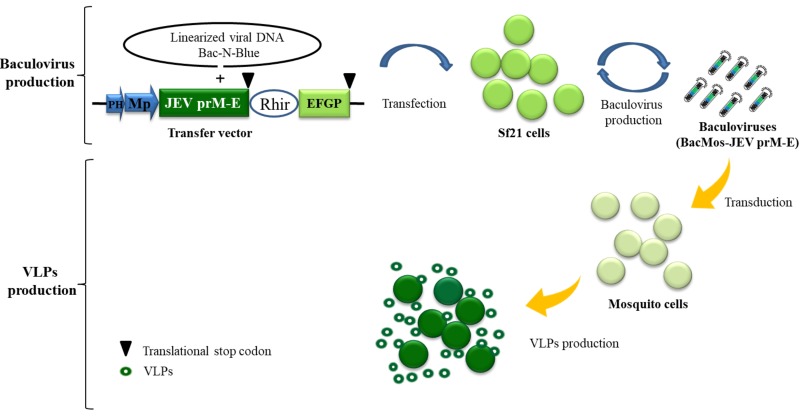
Schematic illustration of baculovirus/mosquito (BacMos) for Japanese encephalitis (JE) virus-like particles (VLPs) production. A recombinant baculovirus bearing a DNA casttest of PH-Mp-JEV precursor membrane protein and envelope (prME)-Rhir-EGFP generated by a Bac-N-Blue system are produced from Sf21 cell. Mosquito cells are efficiently transduced by baculoviruses (BacMos-JEV-prME), and JEV prME expression and VLP (hollow green circles) secretion were induced. Abbreviations: PH, polyhedrin promoter; Mp: Mosquito promoter (hr1pag1); JEV prME: an insect codon-optimized JEV prME gene (AAB66485, 105-794th amino acid); Rhir, RhPV 5′-UTR IRES; EGFP, enhanced green fluorescent protein gene.

**Figure 2 viruses-12-00336-f002:**
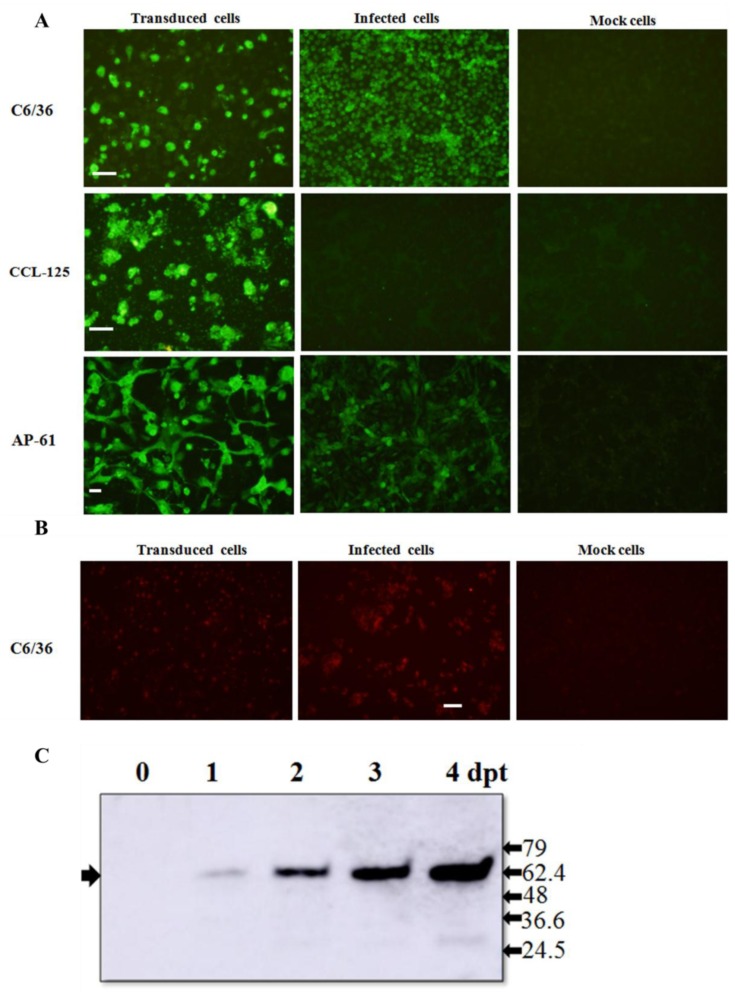
Expression of JEV glycoproteins in BacMos-JEV prME-transduced mosquito cells. Immunofluorescence assays for the detection of JEV E (**A**) and prM (**B**) glycoproteins. Three mosquito cells, C6/36 (**upper panels**), CCL-125 (**center panels**), and AP-61 (**lower panels**), were transduced with BacMos-JEV-prME (**left panels**) or infected with JEV at an MOI of 5 (**center panels**). After 5 days (transduction) or 2 days (infection), BacMos-JEV-prME-transduced cells (**left panels**), JEV-infected cells (**center panels**), and mock cells (**right panels**) were fixed and stained using Mab specific to either JEV E (**A**) or prM (**B**) glycoproteins. Cells were examined using green (**A**) or red (**B**) filters. Scale bar = 100 μm. Western blot analysis results for the detection of JEV E glycoprotein in C6/36 cells transduced using BacMos-JEV-prME (**C**). Total cell lysates were harvested at indicated times (dpt) and subjected to Western blot analysis to detect JEV E glycoproteins (indicated by arrows on the left). Protein sizes (kDa) of markers are shown on the left.

**Figure 3 viruses-12-00336-f003:**
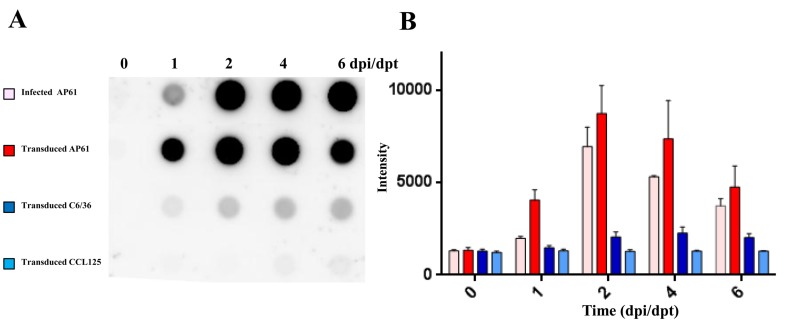
Results of dot blot analysis for the detection of secreted JEV viral glycoproteins. Three mosquito cell lines (C6/36, CCL-125, and AP-61) were transduced with BacMos-JEV-prME at an MOI of 2. C6/36 cells were infected with JEV at an MOI of 0.1. Culture medium was harvested at the indicated dpi and subjected to dot blot analysis (**A**) and quantification (**B**).

**Figure 4 viruses-12-00336-f004:**
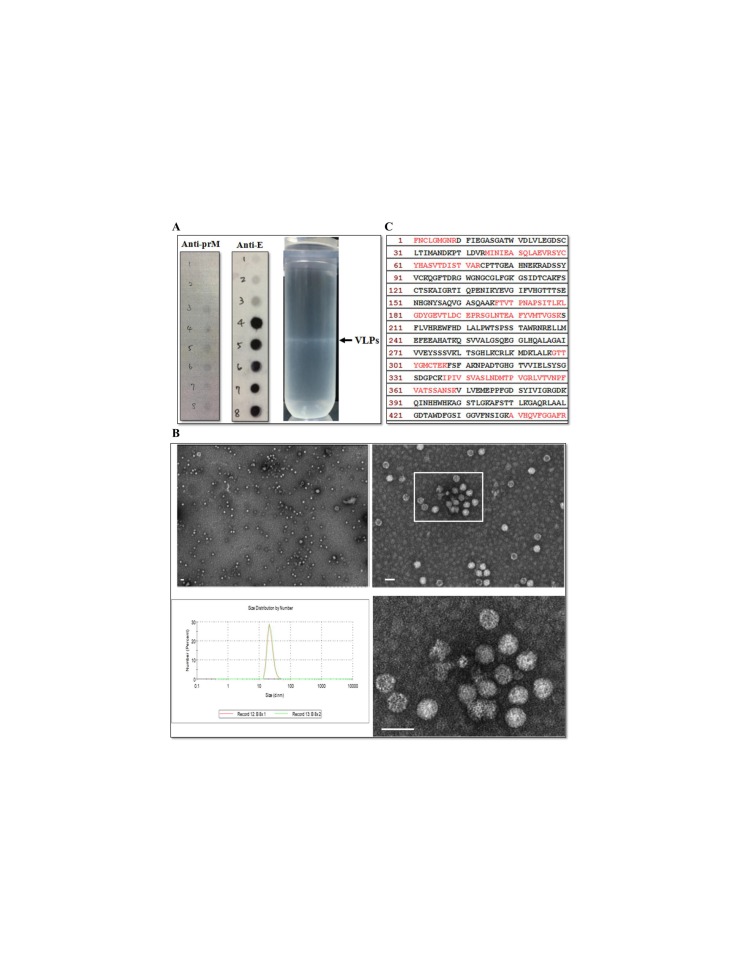
VLPs characterization. (**A**) Sucrose gradient banding JE VLPs. Representative photograph (**left panel**) of the sucrose density gradient following ultracentrifugation to isolate JEV VLPs. Arrow on the left indicates the VLP band. Fractions on the right from 1 (**top**) to 8 (**bottom**) were subjected to dot blot analysis (**right two panels**) using the indicated antibodies. (**B**) Electron micrographs and size distribution of JEV VLPs. Purified JEV VLPs were examined using transmission electron microscopy (**upper** and **lower right panels**). The boxed area in the upper right panel contains the same tissue section as that shown in the bottom panel. Scale bars = 50 μm. The size distribution of JEV VLPs (**lower left panel**) was determined using DLS. (**C**) Identification of peptides from E proteins in VLPs. Partially purified VLPs from fraction 5 were resolved using SDS-PAGE, and the Coomassie blue-stained band corresponding to E proteins was analyzed using mass spectrometry. The amino acid sequences of JEV E glycoproteins are shown, with the matching peptides obtained from the LC–MS/MS analysis shown in red.

**Figure 5 viruses-12-00336-f005:**
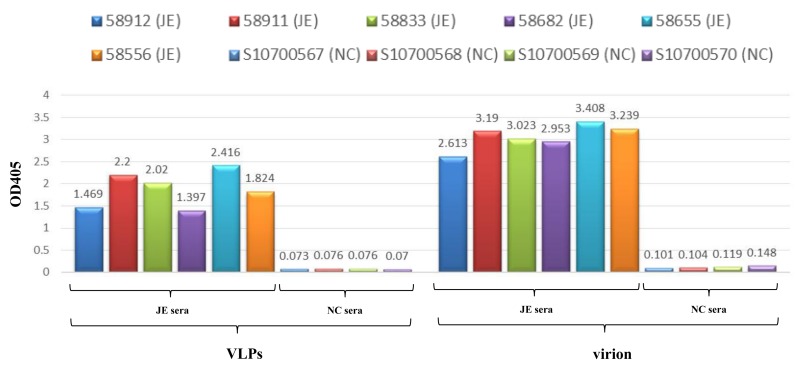
Epitope analysis of JEV VLPs. JEV VLPs (**left**) and virions (**right**) were subjected to MAC-ELISA for human sera, including six sera of JEV infection (JE sera): 58,912 (JE), 58,911 (JE), 58,833 (JE), 58,682 (JE), 58,655 (JE), and 58,556 (JE); as well as four normal sera (NC sera): S10700567 (NC), S10700568 (NC), S10700569 (NC), and S10700570 (NC).

**Figure 6 viruses-12-00336-f006:**
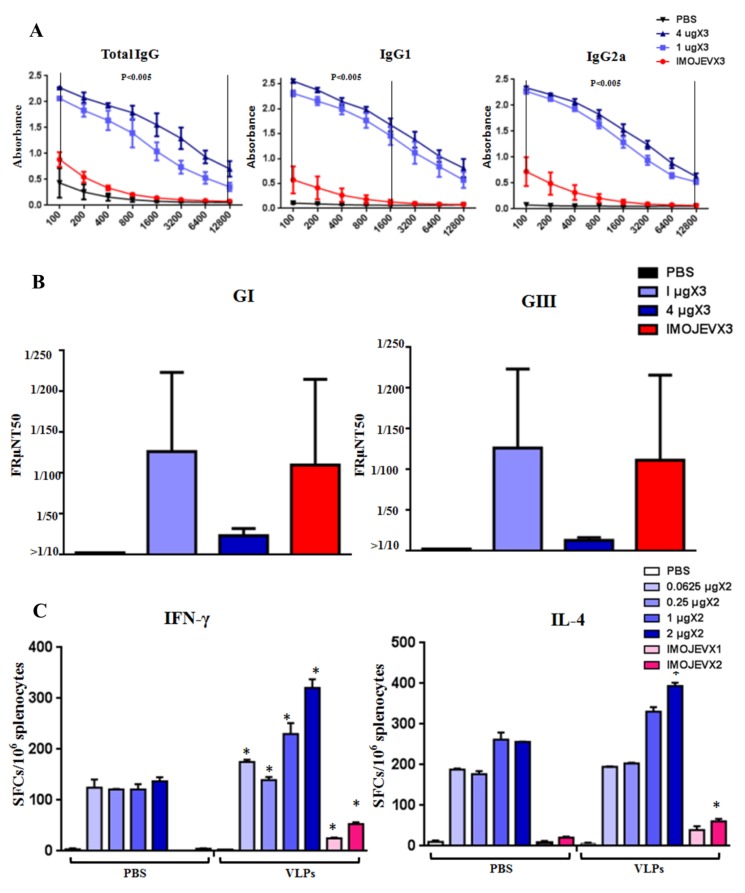
Immunogenicity of mosquito cell-derived VLPs in mice. (**A**) JEV-specific IgG responses. *BALB/c* mice (5 mice/group) received 3 doses of either PBS, 1µg VLPs, 4 µg VLPs, or IMOJEV. The following JEV E-specific antibody responses were determined by ELISA: total IgG (**left panel**), IgG1 (**center panel**), and IgG2a (**right panel**). Error bars indicate the mean SEM of concentrations from specific treatment groups. We identified statistically significant differences (*p* < 0.005) between the group that received 1 µg VLPs and the group that received IMOJEV. (**B**) PRNT_50_ titers against JEV GI and GIII. For both JEV GI and GIII, PRNT_50_ titer values were calculated using 3 mice/group. Error bars indicate the mean SEM of concentrations from individual animals. (**C**) Virus-specific T cell responses. *BALB/c* mice received 2 doses of one of the following: PBS, 0.0625, 0.25, 1, 2 µg VLPs, or IMOJEV. Three months after the second immunization, splenocytes (from 1 mouse/group) were collected and stimulated with purified JEV VLP antigens. Splenocytes secreting either IFN-γ or IL-4 were quantified in triplicate using ELISpot assays. Spot-forming cells (SFCs) were counted and calculated. The data are shown as the means  ±  SEM, and the statistical significance between immunogen stimulations (PBS and VLP) was analyzed using Student’s *t*-test (* *p* < 0.05).

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
