# Peer review of "Mosquito Cell-Derived Japanese Encephalitis Virus-Like Particles Induce Specific Humoral and Cellular Immune Responses in Mice"

_viruses, 2020, doi:10.3390/v12030336_

Round 1

Reviewer 1 Report

The manuscript by Chang and colleagues presents an interesting and comprehensive evaluation of a JEV VLP produced in mosquito cells. The methods appear sound and I was impressed with the detailed characterization of the VLPs that was presented. Comments for improvement of the manuscript, all of which are trivial, include:

Line 26-27: The phrase “This effectively neutralizes antibody responses to JEV GI and GIII ..” seems confusing. Would it be better to state something like “This response effectively neutralized both JEV GI and GIII ..”

Line 60-61: If correct, I suggest changing “A mosquito-derived dengue type 2 VLPs from stable C6/36 clones has been ..” to “Mosquito cell-derived dengue 2 VLPs produced from stable C6/36 cell clones ..”

Line 76: Antibiotic-Antimycotic. -> antibiotic-antimycotic solution.

Line 152: Suggest “Following this, the plates were blocked using ..”

Missing superscripts for powers of 10: line (1 × 107) and line

Line 176: The statement “Two other groups of mice were SC inoculated with either IMOJEV or PBS.” was already made just a few lines above and seems redundant.

Line 198: complements -> complement

Line 203: “that yielded a PRNT50.” -> that reduced virus titer by 50%.

Line 219: “This study developed an efficient ..” -> “We developed ..” (the study did not develop anything)

Line 293: “specific binding activity to sera from” -> binding to immunoglobulins in sera

Line 393: I think you should have a reference to the statement “However, IMOJEV poses risks to pregnant and breastfeeding women as well as to individuals with immune deficiencies.”

Author Response

Thank you for drawing our attention to these issues. We have revised sentences to read as follows:

  1. Line 26-27: The phrase “This effectively neutralizes antibody responses to JEV GI and GIII ..” seems confusing. Would it be better to state something like “This response effectively neutralized both JEV GI and GIII ..”

Reply: “This response effectively neutralized both JEV GI and GIII and elicits a mixed Th1/Th2 response in mice.”

  1. Line 60-61: If correct, I suggest changing “A mosquito-derived dengue type 2 VLPs from stable C6/36 clones has been ..” to “Mosquito cell-derived dengue 2 VLPs produced from stable C6/36 cell clones ..”

Reply:“Mosquito cell-derived dengue 2 VLPs produced from stable C6/36 cell clones has been characterized in immunogenicity [35].”

Line 76: Antibiotic-Antimycotic. -> antibiotic-antimycotic solution.

Reply:“Mosquito cell lines CCL-125 (Aedes aegypti) and C6/36 (Aedes albopictus) were cultured in RPMI1640 medium (GIBCO, Invitrogen, CA) containing 10 % fetal bovine serum and 1x antibiotic-antimycotic solution (GIBCO, Invitrogen, CA).”

  1. Line 152: Suggest “Following this, the plates were blocked using ..”Missing superscripts for powers of 10: line (1 × 107) and line

 Reply:“Following this, plates were blocked using blocking buffer (PBS and 1% BSA), and 100-fold dilutions of human sera (six sera of JEV infection: 58912, 58911, 58833, 58682, 58655, and 58556; four normal sera: S10700567, S10700568, S10700569, and S10700570) were added to the wells. Wells were subsequently incubated at 37 °C for 30 min. Following this, free antibodies were washed out, 100 μl of JE VLPs from the culture supernatant of BacMos-JEV-prME-transduced AP-61 cells or virions of JEV JaGAr-01 strain (1X107 PFU/ml) was added to each well, and all wells were incubated at 37 °C for another 30 min.”

  1. Line 176: The statement “Two other groups of mice were SC inoculated with either IMOJEV or PBS.” was already made just a few lines above and seems redundant.

Reply: Two other groups of mice were SC inoculated with either IMOJEV or PBS. The sentence has been deleted.

  1. Line 198: complements -> complement

Reply: “Following the heat inactivation of complement, 100 μl of the serial dilutions were incubated with an equal volume of JEV solution containing approximately 100 PFU of JEV I or JEV III.”

  1. Line 203: “that yielded a PRNT50.” -> that reduced virus titer by 50%.

Reply: “After culturing, plaques were stained and counted. Nabs titers were defined as the reciprocal of the maximum dilution of serum that reduced virus titer by 50%.”

  1. Line 219: “This study developed an efficient ..” -> “We developed ..” (the study did not develop anything) Reply: “We developed an efficient gene delivery method to facilitate the production of JEV VLPs in mosquito cells.”

  1. Line 293: “specific binding activity to sera from” -> binding to immunoglobulins in sera Reply: “MAC-ELISA results (Figure 5) revealed that VLPs and virions exhibited (1) binding to immunoglobulins in sera all JEV patients and (2) minor binding activity to normal sera.”

  1. Line 393: I think you should have a reference to the statement “However, IMOJEV poses risks to pregnant and breastfeeding women as well as to individuals with immune deficiencies.” Reply: “However, IMOJEV is not recommend to do vaccination for pregnant and breastfeeding women as well as immunocompromised persons [58].”

Reviewer 2 Report

At present, inactivated whole virus  or live attenuated vaccines are used for prevention of JEV infection, which may have a limited efficacy, depending on the virus strain circulating in particular region, and undesirable side effects. Therefore, improvement of JEV vaccine is warranted. VLPs , as an safe, easy-to- produce and well immunogenic form of subunit vaccines, had been intensively studied in several virus infections, including JEV. Originality of this paper lies in an usage of  mosquito cell expression system, which seems to have some advantages above the insect cells used so far. Production and purification of VLPs in mosquito cells is well described. Immunogenicity of the VLPs is confirmed by several approaches including an experiment on  the animals and seems to be even higher, when compared with an licensed vaccine IMOJEV. These results implicate mosquito cells- produced JEV-derived VLPs as potential candidate for JEV subunit vaccine.

  • For the purpose of potential human vaccine, the cross-protection properties of  JEV GIII-derived VLPs gainst various JEV genotypes should be more extensively studied: in the paper  cross – neutralization of JEV GI is documented, but no data are present as for other JEV genotypes, especially GV.
  • In mice VLPs elicited both humoral and cellular immune response, but, in contrast to the total IgG antibody and T-cell response, neutralising antibodies were not produced in dose-dependent manner. The authors should explain this anomaly in more detail.

Minor comments:

  • In description of ELISA assay for evaluation IgG response ( M+M, par. 2.10) a final step of the  color reaction stopping is omitted. In this setting, the colorimetric reaction will not run in standard conditions.
  • In Results (Par.3, line 236)…”three transduced cell lines” should be used instead of “ three transduced cells”
  • In Discussion ( Par. 4 , line 411): T cells are not responsible for phagocytosis but for cytolytic destruction of virus-infected cells.

Author Response

  1. For the purpose of potential human accine, the cross-protection properties of JEV GIII-derived VLPs gainst various JEV genotypes should be more extensively studied: in the paper cross – neutralization of JEV GI is documented, but no data are present as for other JEV genotypes, especially GV.

 Reply: Thank you for drawing our attention to this interesting point.

JEV GV is non-available in our institute. Instead of this, mosquito-derived JEV GV VLP will be generated using BacMos, and JEV GV VLP-induced neutralizing activity against JEV GI and III will be evaluated.

  1. In mice VLPs elicited both humoral and cellular immune response, but, in contrast to the total IgG antibody and T-cell response, neutralising antibodies were not produced in dose-dependent manner. The authors should explain this anomaly in more detail.

Reply: Thank you for drawing our attention to this important issue. we have explained non-dose-dependent manner in IgG, T-cell, and Nabs responses in discussion section,  as follows:   

“However, there are no statistic differences in E-specific IgG response (1 vs 4 µg doses) (Figure 6 A). That indicates immunization of 1 µg dose could be sufficient for inducing a plateau IgG response. There are also no statistic differences in Nabs (1 vs 4 µg doses) (Figure 6 B) and Th1/Th2 (1 vs 2 µg doses) (Figure 6 C) responses, which could be resulted in part of low number of mice employed. However, a 1:10 dilution in a 50%PRNT considered to be indicative of protection from JEV infection [70]. Notably, the 1 μg dose should be sufficient to maximize the Nabs reaction in mice. It is need to close evaluate JE VLP immunogenicity on dosage using adequate number of mice in future.”

Minor comments:

  1. In description of ELISA assay for evaluation IgG response ( M+M, par. 2.10) a final step of the color reaction stopping is omitted. In this setting, the colorimetric reaction will not run in standard conditions.

Reply: Thank you for bring this to our attention. We did not add the stop solution, but have a fixed the reaction time to control the color change. It will be performed according to standard conditions in our future experiments.

  1. In Results (Par.3, line 236)…”three transduced cell lines” should be used instead of “ three transduced cells”

Reply: We have revised sentences to read:

“Dot blot analysis results (Figure 3A) showed that three transduced cell lines and one infected cell had secreted JEV E glycoproteins.”

  1. In Discussion ( Par. 4 , line 411): T cells are not responsible for phagocytosis but for cytolytic destruction of virus-infected cells.

Reply: We have revised sentences to read:

“The enhanced IFN-γ and IL-4 induced secretion of splenocytes is associated with the functional cytotoxic T cells which are responsible for cytolytic destruction of virus-infected cells.”